# Plasma-Treated Solutions (PTS) in Cancer Therapy

**DOI:** 10.3390/cancers13071737

**Published:** 2021-04-06

**Authors:** Hiromasa Tanaka, Sander Bekeschus, Dayun Yan, Masaru Hori, Michael Keidar, Mounir Laroussi

**Affiliations:** 1Center for Low-Temperature Plasma Sciences, Nagoya University, Furo-cho, Chikusa-ku, Nagoya 464-8601, Japan; hori@nuee.nagoya-u.ac.jp; 2ZIK Plasmatis, Leibniz Institute for Plasma Science and Technology (INP), Felix-Hausdorff-Str. 2, 17489 Greifswald, Germany; 3Department of Mechanical and Aerospace Engineering, The George Washington University, Washington, DC 20052, USA; keidar@gwu.edu; 4Plasma Engineering and Medicine Institute, Old Dominion University, Norfolk, VA 23508, USA; mlarouss@odu.edu

**Keywords:** cold physical plasma, low-temperature plasma, nonthermal plasma, oncology, PAM, plasma-activated medium, plasma medicine, reactive oxygen species, reactive nitrogen species

## Abstract

**Simple Summary:**

Cold physical plasma is a partially ionized gas generating various reactive oxygen and nitrogen species (ROS/RNS) simultaneously. ROS/RNS have therapeutic effects when applied to cells and tissues either directly from the plasma or via exposure to solutions that have been treated beforehand using plasma processes. This review addresses the challenges and opportunities of plasma-treated solutions (PTSs) for cancer treatment.

**Abstract:**

Cold physical plasma is a partially ionized gas generating various reactive oxygen and nitrogen species (ROS/RNS) simultaneously. ROS/RNS have therapeutic effects when applied to cells and tissues either directly from the plasma or via exposure to solutions that have been treated beforehand using plasma processes. This review addresses the challenges and opportunities of plasma-treated solutions (PTSs) for cancer treatment. These PTSs include plasma-treated cell culture media in experimental research as well as clinically approved solutions such as saline and Ringer’s lactate, which, in principle, already qualify for testing in therapeutic settings. Several types of cancers were found to succumb to the toxic action of PTSs, suggesting a broad mechanism of action based on the tumor-toxic activity of ROS/RNS stored in these solutions. Moreover, it is indicated that the PTS has immuno-stimulatory properties. Two different routes of application are currently envisaged in the clinical setting. One is direct injection into the bulk tumor, and the other is lavage in patients suffering from peritoneal carcinomatosis adjuvant to standard chemotherapy. While many promising results have been achieved so far, several obstacles, such as the standardized generation of large volumes of sterile PTS, remain to be addressed.

## 1. Introduction

Cancer is a devastating condition, and the second leading cause of death in Western countries [1]. Despite ongoing improvements in cancer therapy, novel research lines are warranted to improve the clinical efficacy of cancer treatments. A recent development in oncology is increasingly focusing on combination therapies that tackle tumor cells via several mechanisms, either simultaneously or consecutively. While many therapeutic approaches focus on targeted therapies and immuno-therapies, promising research motivated continuously tackling tumors on a much broader scale involving cancer metabolism and reactive oxygen and nitrogen species (ROS/RNS) and their associated redox signaling pathways [2,3,4].

Cold physical plasma, which also goes by low-temperature plasma or nonthermal plasma, is a partially ionized gas generating a multitude of ROS/RNS simultaneously. More than two decades ago, it was proposed that these ROS/RNS can be used for therapeutic purposes. Early work focused on plasma-based disinfection of biologically contaminated surfaces and liquids [5,6], which was later transferred to the medical field and wound decontamination [7,8]. In addition to decontamination, it was soon realized that plasma-derived ROS/RNS also target eukaryotic cells by cell-intrinsic mechanisms [9,10]. Moreover, a recent randomized clinical trial found that the disinfection properties of plasma treatment are negligible in promoting wound healing in a cohort of diabetic patients [11]. As ROS/RNS follow hormetic responses in biology [12], it seems natural to investigate exaggerated ROS/RNS exposure from cold physical plasmas as anticancer agents. Plasma-based cancer research started more than a decade ago with promising in vitro [13,14,15] and in vivo studies [16,17] and, ever since, has generated significant interest in experimental oncology [18,19] with the first promising results being obtained in palliative cancer patients in recent years [20,21].

While the direct application of cold physical plasma to cells and tissues is promising and approved in European dermatology centers [22], an increasing need has emerged to utilize the plasma-derived ROS/RNS more flexibly as putative injections. One way to achieve this is the treatment of liquids or solutions with plasma devices [23,24]. In this process, the plasma-derived ROS/RNS are delivered from the plasma gas phase into the liquid phase (Figure 1). As most ROS/RNS are short-lived by virtue of their nature, this comes at the expense of their deterioration [25], yet leaving a delicate mixture of long-lived ROS/RNS that might even recombine or react with short-lived species again when in contact with cells [26]. Since the idea of investigating these liquids is a future clinical application, two routes have been proposed so far. The first involves the injection into the bulk tumor, while the second relates to the lavage of the peritoneal cavity in the disease of disseminated peritoneal carcinomatosis, currently tackled with HIPEC and PIPEC therapy [27].

## 2. Terminology

Such treated liquids go by several terms, such as plasma-activated medium (PAM) [29], plasma-activated solution (PAS) [30], plasma-stimulated medium (PSM) [31], plasma-activated Ringer’s lactate (PAL) [32], plasma-treated liquid (PTL) [33], plasma-oxidized liquid (POL) [27], plasma-treated medium (PTM) [34], plasma-activated water (PAW) [35], and plasma-activated acetic acid Ringer’s solution (PAA) as well as plasma-activated bicarbonate Ringer’s solution (PAB) [36]. Here, we propose harmonizing the nomenclature in the field of plasma medicine by using the term plasma-treated solutions (PTSs) for the following reasons. First, the term “solution” per definitionem includes all types of liquids that have dissolved organic or inorganic compounds, which is the case for all types of solutions used in experimental research as well as clinically accredited solutions [37]. Even though this is not entirely the case for plasma-treated water, the plasma treatment submerges reactive components generated in the ambient air such as nitrite and nitrate into the liquid, effectively generating a solution as well. Second, the term “treated” is a neutral expression describing that a plasma process modified the solution. Hence, every solution exposed to plasma is treated but not necessarily activated, as termed in many previous studies. Activated implies a biomedical response after exposure to such a solution. However, in the case of very short plasma treatment times or low energy, this is not necessarily the case, while at the biochemical level, trace amounts of modified organic or inorganic solvents and ROS/RNS might still be detectable. Third, the term medium should be avoided. Cell culture media differ to extreme extents, differences in inactivation properties with the same treatment times of about 10× were observed [38], discouraging a general utilization of the term “medium”. By contrast, we encourage specifying the type of solution immediately behind PTS—for instance, PTS-RL (Ringer’s lactate) or PTS-saline (sodium chloride). When using cell culture media PTSs, the exact chemical composition of the type of media should be given in the methods section along with the manufacturer and part number. This is the first important step of standardizing reports projected on this topic. Along similar lines, each study should contain a basic characterization of the essential types of ROS/RNS generated in the solution by the plasma source, together with information on evaporation (and its compensation) and changes in the pH. After all, the exact biochemical description of PTSs is vital in understanding and optimizing this approach towards clinical translation.

## 3. In Vitro Experiments of Plasma-Treated Solutions (PTSs) for Cancer Treatment

It has been reported that PTSs selectively kill glioblastoma brain tumor cells against astrocyte normal cells [23]. Antitumor effects of PAM have been demonstrated—for instance, in ovarian cancer cells [39], colon cancer cells [37,40], lung cancer cells [41,42], gastric cancer cells [43], pancreatic cancer cells [44,45,46], leukemia cells [47,48], melanoma cells [49], and squamous cell carcinoma [50]. Antitumor effects of plasma-treated Ringer’s lactate solution, plasma-treated acetic acid Ringer’s solution, and plasma-treated bicarbonate Ringer’s solution were investigated, and interestingly, the lactate and acetic acid solutions effectively killed ovarian cancer cells. In contrast, the plasma-treated bicarbonate solution did not [36]. These results suggest that plasma-treated lactate and acetate are important products for obtaining antitumor effects for these solutions.

For cell culture media, reaction products are complex since they contain about 30 components. However, several components such as amino acids [31] and fetal bovine/calf serum, as well as other compounds such as pyruvate [51], have been investigated for their individual contributions to cytotoxic effects. In general, the drawback of using cell culture media is their complex composition, which disallows unambiguously identifying single or small groups of compounds mainly responsible for the toxicity observed. Moreover, there are many different types of standard cell culture mediums, such as RPMI and DMEM, and unfortunately, many of the differences relate to antioxidants as additives—e.g., pyruvate. If not appropriately referenced, this disallows comparison of the results obtained with plasma-treated media in the field. A comprehensive study comparing several types of media is lacking so far. One study comparing IMDM and RPMI media found a 10-fold difference in mediating cytotoxic effects [38], showing only the tip of the iceberg of how the interpretation of results across several studies may be challenging. Less drastic but still potent differences were recently suggested [52]. In terms of clinical translation, clinically approved solutions seem more promising for apparent reasons. Their composition is defined apart from standardized and quality-controlled production processes needed for medical application. In a previous study, we compared six clinically approved solutions (Table 1) and their storability for their suitability as clinically relevant PTS compared to RPMI cell culture media and phosphate-buffered saline (PBS) [37]. We found sodium chloride and Ringer’s lactate to be especially promising agents for future research along more clinically relevant avenues.

### 3.1. Reactive Species in Plasma-Treated Solutions (PTSs)

Cold physical plasma interacts with oxygen, nitrogen, and water in the air and generates short-lived reactive species in the gas phase such as nitric oxide, ozone, hydroxyl radicals, singlet oxygen, and superoxide anion [53]. Long-lived species such as hydrogen peroxide, nitrite, and nitrate are major components in PTSs [25]. High concentrations of hydrogen peroxide are known to be cytotoxic [54]. For several years, Georg Bauer proposed a synergy between the different PTS components—namely, hydrogen peroxide and nitrite, especially when interacting with enzymes that have located a tumor cell membrane [55,56,57]. Peroxynitrite is produced from hydrogen peroxide and nitrite, followed by the primary singlet oxygen. It has been reported that the primary singlet oxygen causes inactivation of membrane-associated catalase, and hydrogen peroxide and peroxynitrite are produced continuously at the site of locally inactivated catalase, which leads to the generation of secondary singlet oxygen [26]. Other reactive species derived from solutes contribute to physiological responses in PTS-exposed cells (Figure 2, [29]). For example, lactate in Ringer’s lactate solution is a crucial antitumor component when treated with cold physical plasma, and NMR analyses revealed that acetyl- and pyruvic acid-like groups are generated in PTSs (Ringer’s lactate) [36].

The specific cellular response in the nutrient-starved environment and the nutrient-rich environment may cause cancer cells to show drastically different responses to extracellular reactive species such as H_2_O_2_ [58]. The cytotoxicity of reactive species on mammalian cells was found to be dampened in one study when the PTS was made up of simple buffered solutions such as PBS [36,59]. Nevertheless, ROS/RNS are vital components in PTS. These can be monitored in living cells in vitro using fluorescent redox-sensitive reporter probes that can be analyzed by flow cytometry [54], microscopy [60], or microplate readers. The drawback of these probes is the lack of specificity towards individual types of ROS/RNS once entering the intracellular compartment, as reported many times [61,62]. Notwithstanding, they prove useful in estimating intracellular redox changes with tendencies towards some types of ROS over others. For instance, intracellular ROS are often detected using 5-(and-6)-chloromethyl- 2′,7′-dichlorodihydrofluorescein diacetate, acetyl ester (CM-H_2_DCFDA), although it has been reported that its fluorescent product DCF is not a direct consequence of ROS/RNS oxidation but rather an enzymatic product of intracellular oxidase. Direct plasma treatment and PTSs often induce intracellular ROS on cells, and the extent of this finding for PTS exposure depends on the type of solution used. For example, plasma-treated cell culture media were previously suggested to have stronger DCF signals compared to plasma-treated Ringer’s lactate in U251SP cells [63]. To obtain more details of intracellular ROS dynamics, 3′-(p-aminophenyl) fluorescein (APF) and 3′-(p-hydroxyphenyl) fluorescein (HPF) are frequently used probes in plasma medicine [46,54,60,64,65]. Intracellular hydroxyl radicals, peroxynitrite, and OCl^-^ are detected using APF, while intracellular hydroxyl radicals and peroxynitrite are detected using HPF. In addition to these dyes, intracellular hydrogen peroxide, nitric oxide, peroxynitrite, superoxide anion, and OCl^-^ can be detected using other reagents in PTS-exposed HeLa cells to dissect putative intracellular ROS/RNS following exposure (Figure 3) [60]). Up to 2 h after exposure, intracellular hydrogen peroxide, nitric oxide, and superoxide anion were dominant. Intracellular peroxynitrite was negligible. After 5 h, intracellular hydrogen peroxide, nitric oxide, and superoxide anion decreased, and intracellular peroxynitrite increased. While these data are interesting, it nevertheless needs to be noted that direct plasma exposure was found to be significantly more toxic in U251 tumor spheroids compared to PTS (PBS) treatment [66].

### 3.2. Factors Affecting the Anticancer Efficacy of Plasma-Treated Solutions (PTSs)

The carriers of these reactive species, such as cell culture media, PBS, and other solutions such as lactate solutions, are important factors affecting the anticancer efficacy of PTSs. Specific cancer cell lines may be vulnerable to one PTS, such as medium, compared to another PTS, such as PBS, as seen for the pancreatic tumor cells PA-TU-8988T cells and glioblastoma cells U87MG [58] (Figure 4). PTS can also be combined with chemotherapy affecting the efficacy of PTS. Using four different human pancreatic cancer cell lines and the clinically relevant drugs to target those cells, cisplatin and gemcitabine, additive toxicity with PTS was found (Ringer’s lactate, kINPen argon plasma jet) in both 2D cultures and three-dimensional multicellular tumors grown on the CAM of chicken embryos (in ovo). [46]. A synergistic action of PTS (fully supplemented RPMI, kINPen argon plasma jet) and gemcitabine was also confirmed in murine pancreatic cancer cells and found to be selective compared to primary murine fibroblasts [67]. However, it should be noted that such treatment also severely affects the viability of human immune cells, such as monocytes and T cells [68], while leaving differentiated macrophages largely unaffected [69].

### 3.3. Intracellular Molecular Mechanism of Cancer Cell Death Induced by Plasma-Treated Solution (PTS)

The transmembrane diffusion of reactive species may affect the cytotoxicity of PTS (cell culture medium) on cancer cells. One example is the study on aquaporins (AQPs) [70]. Knocking down of the gene encoding AQP8, one type of H_2_O_2_ channel in glioblastoma cells, reduced the cytotoxicity of PTS.

To identify intracellular molecular mechanisms of cell death of PTS-exposed cells, many cell lines have been used. For example, U251SP, U87MG, LN229, and T98G are glioblastoma cell lines, and their genetic backgrounds are different. The *TP53* gene is the wildtype in the U87MG cell line, and the *PTEN* gene is the wildtype in the LN229 cell line. Sensitivity against PTS is different among those cell lines [71]. Further studies are needed to elucidate what the effectiveness of specific types of PTS depends on. The ovarian cancer cell lines SKOV3, ES2, NOS2, and TOV21G have also been investigated [72]. Ovarian clear-cell carcinoma (CCC) is a subtype of epithelial ovarian carcinoma (EOC), and it is naturally chemoresistant and often associated with a poor prognosis. TOV21G was used as a CCC cell line.

Direct plasma exposure and PTS induce apoptosis in cells. For example, cleaved Caspases 3 and 7 were detected in PTS-exposed (cell culture medium) U251SP cells [23] and PTS-exposed (Ringer’s lactate) U251SP cells [36]. However, intracellular molecular mechanisms of cell death are different between both exposure types [63]. Transcriptome microarray analyses of PTS-exposed U251SP suggested that PTS-induced gene expression related to GADD45 signaling is activated by oxidative stress and induces apoptosis. The dynamics of gene expression analyses related to GADD45 signaling between PTS- (medium) and PTS-exposed (Ringer’s lactate) U251SP cells revealed that the latter only showed minimal effects in GADD45 signaling (Figure 5, [63]). These results are consistent with the results that PTS (medium) induced more ROS than PTS (Ringer’s lactate). Survival and proliferation signaling pathways such as the PI3K–AKT pathway and the RAS–MAPK pathway were downregulated in PTS-exposed U251SP cells [73]. These results are also consistent with the fact that AKT inhibition induces *GADD45α* gene expression. GAD45α/p38 signaling pathway increases the activation of ATF3 and c-JUN transcription factors, which form the AP-1 complex.

PTS (medium) also induces caspase-independent cell death in A549 nonsmall lung cancer cells [41]. Injured mitochondria but not caspase 3/7 activation were observed in PTS-exposed A549 cells. Bcl_2_ mRNA was significantly decreased and CHOP mRNA was induced in PTS-exposed A549 cells. The formation of poly ADPR was detected in the nuclei of PTS-exposed A549 cells, and the accumulation of AIF surrounding the nucleus was detected in PTS-exposed A549 cells. PTS exposure elevated intracellular calcium ion concentration by activating the nonselective cation channel TRPM2 in A549 cells. Based on these results, intracellular molecular mechanisms of cell death have been proposed (Figure 6, [41]).

PTS affects gene expression, signal transduction, and metabolic networks (Figure 7). The interface between PTS and the cell is the cell membrane, and PTS first affects proteins such as receptors, ion channels, transporters, and lipids on the cell membrane. Then, PTS-affected proteins and lipids alter the signal transduction network. Such alterations affect gene expression and metabolic networks. Systematic analyses such as transcriptomics, proteomics, and metabolomics are powerful tools to understand intracellular molecular mechanisms of PTS-exposed cells.

Moreover, PTS (fully supplemented cell culture medium, kINPen argon plasma jet) was shown to affect mitochondrial dynamics in melanoma cells, critically synergizing with complex I inhibitors to augment cancer cell death (Figure 8) [74].

### 3.4. Some Guidelines to Make PTS

Extending the plasma treatment time (dose) is the most straightforward strategy to enhance the anticancer efficacy of PTSs [23,36,75,76,77]. Several general principles to improve the efficacy of PTS (cell culture medium) have been demonstrated by regulating three treatment factors: the gap between the plasma source and the solution surface, the contacting area between bulk plasma and the surface (Figure 9), and the volume of solution. These principles have recently been tested in other PTSs, such as PBS by other groups [78]. A larger containing area between the bulk plasma and the solution surface will result in a higher reactive species concentration in the PTS [75]. For example, the PTS made in a well of a 6-well plate is much more toxic to cancer cells than the PTS made in a well of a 96-well plate. Additionally, a shorter gap between the plasma source and the solution also generates a more toxic PTS with a higher reactive species concentration. The stronger capacity of PTS can also be achieved by decreasing the solution volume. Furthermore, the basic operational parameters of plasma can also modulate the efficacy of PTSs. For instance, the discharge voltage can noticeably affect the H_2_O_2_ generation in PTS, which further affects the anticancer performance [79].

It is necessary to emphasize that some media components are very reactive to the plasma-derived ROS/RNS. For example, fetal bovine/calf serum (FBS/FCS), a standard component of a typical cell culture media, is highly reactive to ROS. FBS/FCS weakens the efficacy of PTS on glioblastoma cells [80]. In addition, pyruvate is also reactive with H_2_O_2_ in PTS [41]. Cysteine is a common component in nearly all standard cell culture media. Among all 20 amino acids, however, cysteine is most reactive with ROS [80]. Shortly, an ideal PTS should avoid containing FBS, pyruvate, and cysteine if maximizing ROS/RNS concentrations is desired.

### 3.5. The Storage of Plasma-Treated Solutions (PTSs)

Most media used for cell culture purposes have ideal storage temperatures ranging between 2 and 8 °C. Therefore, an ideal PTS made of cell culture media should also be stably stored in such a temperature range. However, the degradation of PTSs (cell culture media) after the storage has been investigated across a broader range of temperatures. The anticancer species and corresponding efficacy of such PTSs will gradually degrade during the storage over a wide temperature range for less than just 1 day [41,75,80,81,82]. For H_2_O_2_, its concentration also drastically decreased after storage, which was just proportional to the decreased efficacy of such PTSs [82]. As a typical example, Mohades et al. observed that a stored PTS (cell culture medium) lost some of its potency in killing cancer cells at 12 h post-PTS exposure (Figure 10) [82]. PTS was stored at room temperature for 1, 8 and 12 h before its use. Comparing the cell viability outcome of SCaBER treated with stored PTS indicated that toxic efficacy of PTS decreased with increasing storage time [82]. However, the plasma exposure time used to generate such PTSs also plays an important role. Reduction in the efficacy of PTS is more significant for shorter compared to longer exposure times, suggesting that ROS/RNS concentrations remain high enough after storage for PTS generated with longer plasma exposure times.

PTS (PBS, kINPen argon plasma jet) was also found to be storable for up to three weeks at −20 °C without a decline of activity [40]. In a structured comparison of 8 different types of solutions, including 6 clinically approved variants, several solutions were disqualified in terms of storage at −20 °C [37]. Sodium chloride (saline) and Ringer’s lactate at 0.9% gave the best results in terms of storability. The temperature of −20 °C was chosen as this is most realistic for a quick on-site-surgery use-on-demand setting.

Without knowing the exact degradation mechanisms, only storage in low-temperature environments (e.g., −80 °C in a freezer or −150 °C in liquid nitrogen) can inhibit PTSs’ degradation over long periods [36,41,81]. For cell culture medium, a degradation mechanism has been recently proposed. Comparing the H_2_O_2_ concentrations in PTS (cell culture medium vs. PBS) after storage at −25, 8 and 22 °C, medium components and some amino acids were found to be responsible for the observations [31]. Cysteine and methionine mainly cause the degradation of H_2_O_2_ PTS (cell culture medium) (Figure 11). The PTS without cysteine and methionine was much more stable than the PTS that included both amino acids during storage at these temperatures [31].

Moreover, it was found that PTS (PBS, kINPen argon plasma jet) could induce apoptosis in human [83] and murine pancreatic cancer cells in vitro and in vivo [45]. Interestingly, murine primary fibroblasts were less affected. The toxic effects on the cancer cells were not affected by adding several amino acids (alanine, leucine, tryptophan, tyrosine, valine; all 20 mM), suggesting their scavenging capability to be minor when added after plasma treatment in this particular setting. By contrast, cysteine added to PBS before plasma treatment did affect cells [84], suggesting that the oxidation generated by the plasma treatment on this amino acid conserves some of the initial toxicity of the short-lived ROS/RNS. Stability and effects of the gap and gas flow rates on ROS/RNS generation in PTS (PBS, kINPen argon plasma) have been investigated previously (Figure 12) [85].

## 4. In Vivo Experiments of Plasma-Treated Solutions (PTSs) for Cancer Treatment

### 4.1. The Anticancer Efficacy of Plasma-Treated Solutions (PTSs)

The effectiveness of PTSs (RPMI cell culture medium) has been tested using xenograft mouse models and animal disease models. For example, chemoresistant ovarian cancer cells (NOS2TR) and its parental ovarian cancer cells (NOS2) were subcutaneously injected into the bilateral flank of mice [39]. The PTS-exposed group of mice received PTS by subcutaneous injection in each side, three times a week. The control group of mice received an untreated medium in the same way. PTS significantly reduced the growth of both parental and chemoresistant ovarian cancer cells (Figure 13, [39]). Similar results were also observed in pancreatic cancer cells.

To examine the effectiveness of the antitumor effects of PTS (cell culture medium), enhanced green fluorescent protein-tagged GCIY gastric cancer (GCIY-EGFP) cell-induced peritoneal carcinomatosis was investigated [78]. The intraperitoneal administration of PTS or untreated medium (control) was performed on days 1 to 4 and days 8 to 11. The in vivo imaging assays were performed on days 1, 8 and 15 and it was demonstrated that PTS treatment inhibited the formation of peritoneal metastases (Figure 14, [78]).

In another study, RPMI cell culture medium (5 mL) was treated with plasma of the clinically accredited argon plasma jet kINPen MED for 10 min. Although a large number of injections were needed (1–2 dozens), an apparent reduction in peritoneal carcinomatosis was identified in a syngeneic orthotopic model of pancreatic cancer using such a PTS (Figure 15) [86]. The model has the benefit that the immune system is not neglected as the tumor cells are of the same genetic background as the mouse strain into which the tumors were inoculated. The findings were accompanied by intratumoral apoptosis (qualitatively and quantitatively shown using TUNEL staining), as well as a decreased intratumoral proliferation (qualitatively and quantitatively showed using Ki67 staining) in the metastatic nodules. PTS exposure led to a significantly improved overall survival of the mice in the PTS group compared to the vehicle medium control. Detailed analysis of the tumor microenvironment (TME) revealed an increased influx of macrophages into the tumor nodules that were repeatedly exposed to PTS in vivo, along with more T cells as well as an increase in calreticulin (CRT) [87], a marker of the immunogenic cancer cell death [88]. In vitro experiments confirmed the selectivity of this approach when comparing the PDA6606 pancreatic cancer cells against murine primary fibroblasts.

In a recent report, we used a PTS (PBS, kINPen argon plasma jet) to investigate its toxicity in four different gastrointestinal tumor cell lines in vitro. In addition to analyzing the liquid chemistry, we observed the onset of ICD in CT26 colorectal cancer cells in vitro. Strikingly, this was accompanied by a massive antitumor action of PTS (NaCl, kINPen argon plasma jet) in vivo and an increase in intratumoral macrophages as well as T cells (Figure 16), which corroborates findings with direct plasma exposure in vivo [89]. This is the first study using a clinically approved solution (NaCl) together with a clinically approved plasma source (kINPen) in a clinically realistic setting (only PTS stored at −20 °C was used to mimic the clinical situation in which PTS likely would be generated before application as the generation time of the PTS of 1 h was too long to be generated ad hoc during surgical procedures) in an immune-competent (syngeneic, immuno-competent) animal model with the tumors growing at the location where they also would appear clinically (orthotopic, peritoneal cavity of metastatic colon cancer) [40].

As concluded from above, the oxidative stress inflicted upon tumor cells using PTSs has an immuno-modulatory dimension. The extent of this dimension is underexplored at the moment. The main idea is that the PTS-induced tumor cell death is immunogenic, thereby promoting the immune system to recognize dying tumor cells in an inflammatory context with subsequent induction and promotion of antitumor immunity. The exact requirements for PTS to induce such an effect remain to be elucidated. However, from the in vivo studies mentioned above, it is clear that PTS exposure of gastrointestinal tumor burden increases influx of professional antigen-presenting cells such as dendritic cells and macrophages. It is also possible that the immune cells present within the TME will be directly by PTS. As mentioned above, T cells severely succumb to oxidation-induced toxicity. Data are intratumoral immune cells and their cell death after PTS exposure has not been reported so far.

Another point is that the anticancer mechanisms of PTSs are of a general nature—it is uncertain as to whether they are more potent in some tumor entities over others. Along similar lines, great heterogeneity is observed within individual tumor entities, which is related to different mutational loads, variation in cell types within the TME, and signaling pathway deregulation. The current response rates to conventional drugs and immunotherapy show that responders and nonresponders can be stratified according to these factors, and it seems likely that PTS therapy will face similar drawbacks. However, progress of the PTS regimen to clinics is absent at the moment, leaving such aspects to be debated in the future. Comparative preclinical studies that incorporate different tumor entities or cell types with different traits of the same entity would greatly facilitate a much-needed gain in the knowledge in the field of PTSs. Such studies should also elaborate on the “dosing” of PTS—i.e., whether hormetic effects are observed that might show subtoxic effects at low doses.

### 4.2. The Safety of Plasma-Treated Solutions (PTSs)

The safety of a PTS (water) in immune-deficient nude mice was investigated using this PTS by oral lavage treatment [35]. The acute toxicity test results showed that the PTS had no lethal effect or other acute toxicity, even when it was made with a 15 min plasma treatment time. After two weeks of exposure, the PTS did not cause significant changes in mice’s body weights and survival status. The major organs, including the heart, liver, spleen, lung and kidney, did not show observable changes. Liver function, renal function, electrolytes, glucose metabolism and lipid metabolism were also not affected by PTS treatment. Only blood neutrophils and mononuclear cells were found to be slightly increased [35].

The repeated application of PTSs (cell culture medium, kINPen argon plasma jet) in the peritoneal cavity of tumor-bearing mice suffering from peritoneal pancreatic carcinomatosis was tolerated well without visible side effects. The relative frequencies of several blood parameters were not affected in the PTS group compared to the vehicle group. This was the case for neutrophils, monocytes, lymphocytes, T helper cells, cytotoxic T cells, B cells, and NK cells. For the non-nucleated blood cells (erythrocytes and thrombocytes), similar counts and volumes were determined in both groups. The quantification of six different chemokines and cytokines from nonstimulated and LPS-stimulated splenocytes of both groups also did not show any differences [86]. Hence, it can be concluded that the repeated application of a PTS (RPMI cell culture medium) does not cause any apparent harm in C57BL/6 mice in vivo.

## 5. Conclusions

Plasma-treated solutions (PTSs) show potent antitumor efficacy in many cancer cell lines in vitro and in vivo. Cutting-edge research currently addresses the immuno-modulating effects of the exposure of PTS to cancer cells and immune cells of the tumor microenvironment. A future step is to increase the translation of this promising field into clinics using solutions accredited for administration in humans. Along those lines, large bulk plasma-generators and ISO-standardized processes are needed to generate sufficient amounts of PTS in a reproducible and quality-controlled manner.

## Figures and Tables

**Figure 1 cancers-13-01737-f001:**
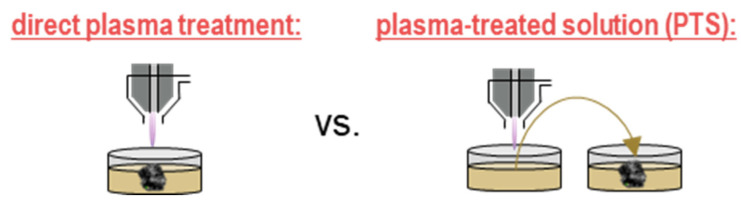
Direct plasma treatment exposes the biological target such as in vitro cells or in vivo or ex vivo tissue directly to the plasma gas phase. Plasma-treated solution (PTS), in turn, is generated by exposing a solution to the plasma gas phase and subsequently transferring this liquid to a target cell culture in vitro or for injection in vivo. Reproduced from [28]. Copyright 2020 MDPI.

**Figure 2 cancers-13-01737-f002:**
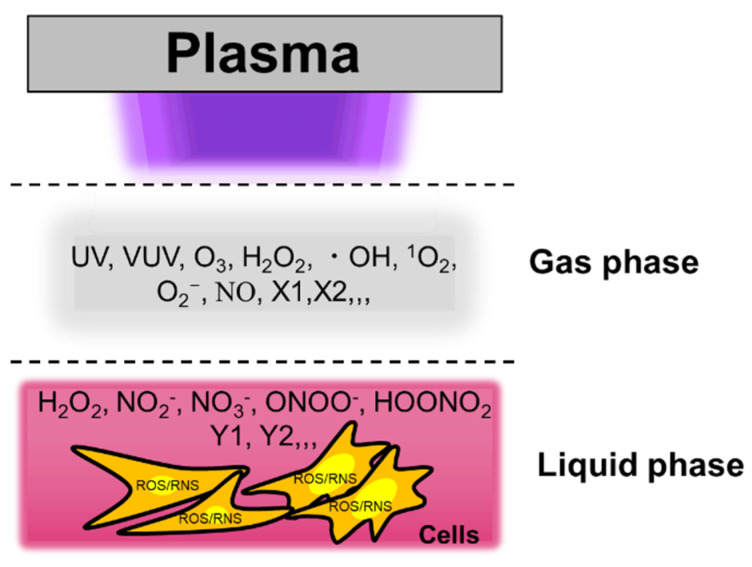
Reactive species in the plasma gas phase dissolve into the bulk liquid to subsequently reach the cells with potentially cytotoxic consequences. The example shows an in vitro setup. Modified from [29]. Copyright 2014 IEEE.

**Figure 3 cancers-13-01737-f003:**
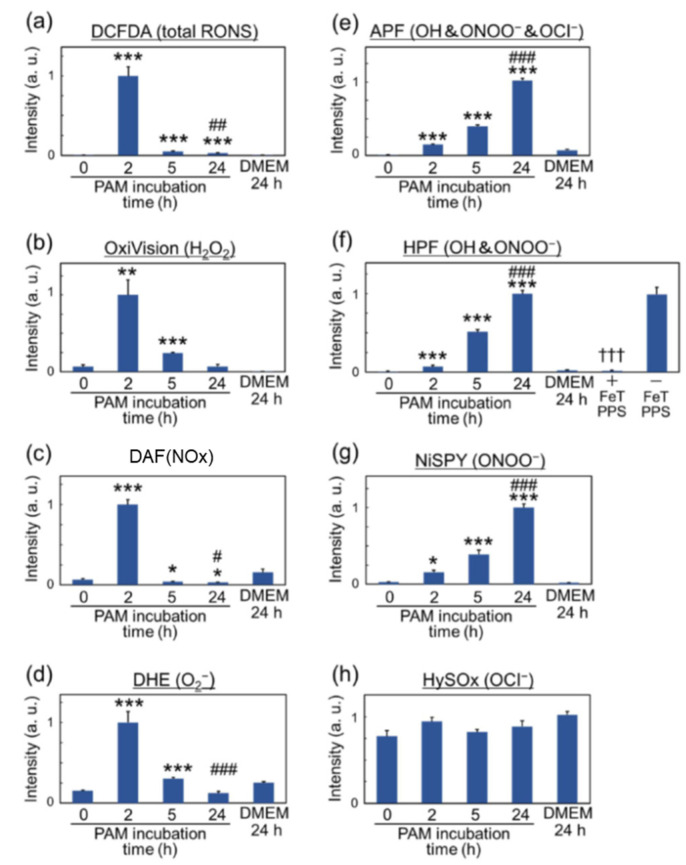
Intracellular reactive oxygen and nitrogen species (ROS/RNS) dynamics in Plasma-Treated Solution (PTS)-exposed HeLa cells. (**a**) total RONS determined using CM-H2DCFDA fluorescent stain reagent; (**b**) H_2_O_2_ determined using OxiVision fluorescent stain reagent; (**c**)•NO determined using DAF-FM-DA fluorescent stain reagent; (**d**) •O_2_^-^ determined using DHE fluorescent stain reagent; (**e**) •OH, ONOO^−^, and OCl^−^ determined using APF fluorescent stain reagent; (**f**) •OH and ONOO^−^ determined using HPF fluorescent stain reagent; (**g**) ONOO^−^ determined using NiSPY-3 fluorescent stain reagent; (**h**) OCl^−^ determined treatment using HySOx fluorescent stain reagent. The symbols * and # indicate significant differences between PAM and nontreatment and between PAM and 24 h DMEM treatment. One symbol; *p* < 0.05, double symbols; *p* < 0.01, triple symbols; *p* < 0.005. Modified from [60]. Copyright 2017 Wiley.

**Figure 4 cancers-13-01737-f004:**
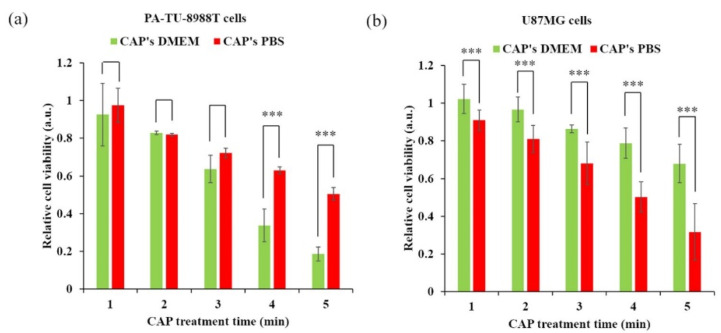
The PTS composition affects its extent of cytotoxicity. (**a**) Pancreatic cancer cell line PA-TU-8988T. (**b**) Glioblastoma cell line U87MG. Student’s *t*-test was performed and the significance is indicated as *** *p* < 0.005. Reproduced from [58]. Copyright 2017 Springer Nature.

**Figure 5 cancers-13-01737-f005:**
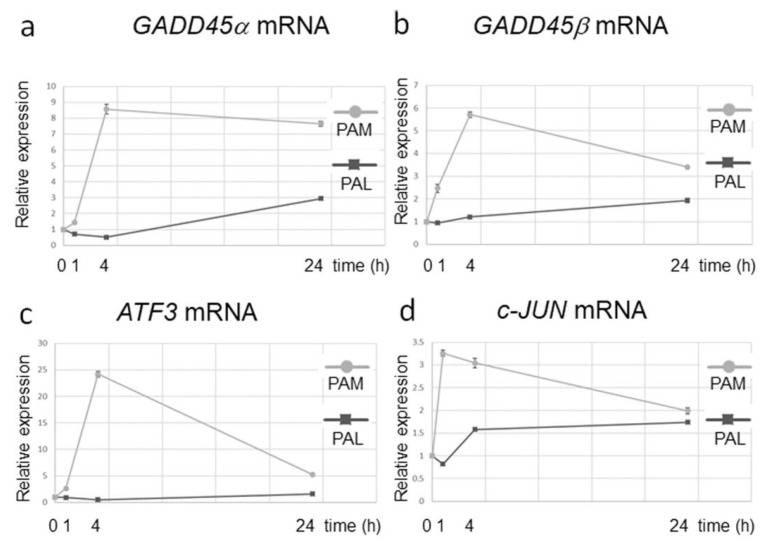
Differences in gene expression dynamics between PTS- (medium, plasma-activated medium (“PAM”)) and PTS-exposed (Ringer’s lactate, plasma-activated Ringer’s lactate (“PAL”)) U251SP cells. Relative mRNA expression of GADD45α (**a**), GADD45β (**b**), ATF3 (**c**), and c-JUN (**d**) was calculated using qRT-PCR. Reproduced from [63].

**Figure 6 cancers-13-01737-f006:**
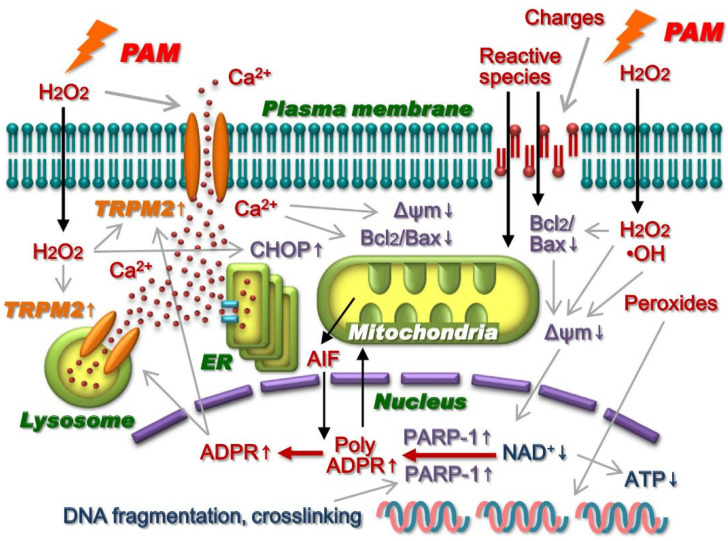
Intracellular molecular mechanisms of cell death in PTS-exposed (cell culture medium) A549 cells. Reproduced from [41]. Copyright 2014 S Elsevier.

**Figure 7 cancers-13-01737-f007:**
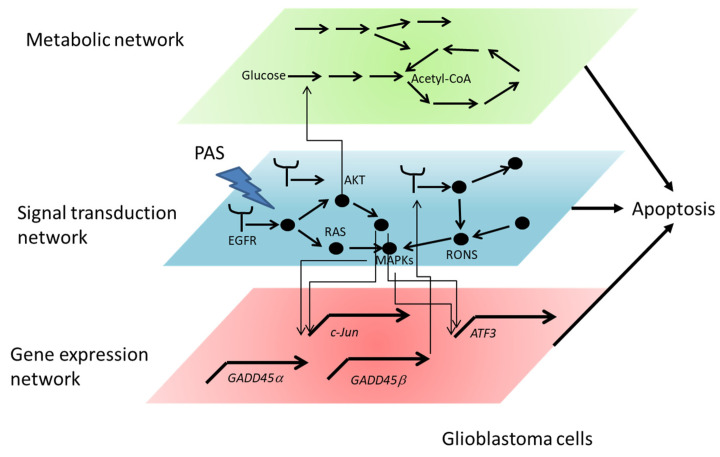
Gene expression network, signal transduction network, and metabolic network affected by PTS.

**Figure 8 cancers-13-01737-f008:**
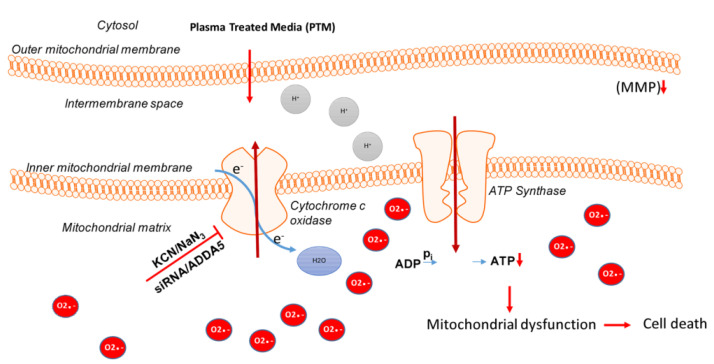
Inhibition of cytochrome C oxidase and the addition of PTS (fully supplemented cell culture medium, kINPen argon plasma jet) leads to an increase in superoxide anions (red) in the mitochondrial matrix that results in loss of MMP and subsequent ATP depletion. This finally leads to an energy crisis and cell death. Reproduced from [74]. Copyright 2018 Springer Nature.

**Figure 9 cancers-13-01737-f009:**
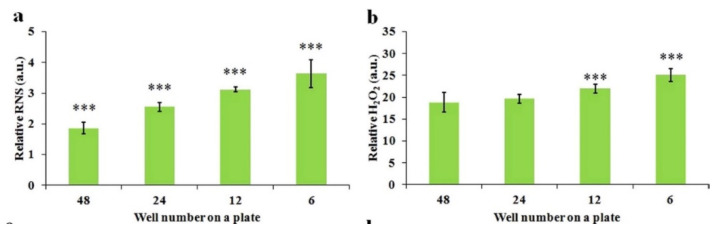
The well size-dependent ROS/RNS accumulation in PTS. (**a**) Relative RNS concentration in 1 mL of PTS. (**b**) Relative H_2_O_2_ concentration in 1 mL of PTS. Student’s *t*-tests were performed and the significance compared with the first bar is indicated as *** *p* < 0.005. Reproduced from [75]. Copyright 2015 Springer Nature.

**Figure 10 cancers-13-01737-f010:**
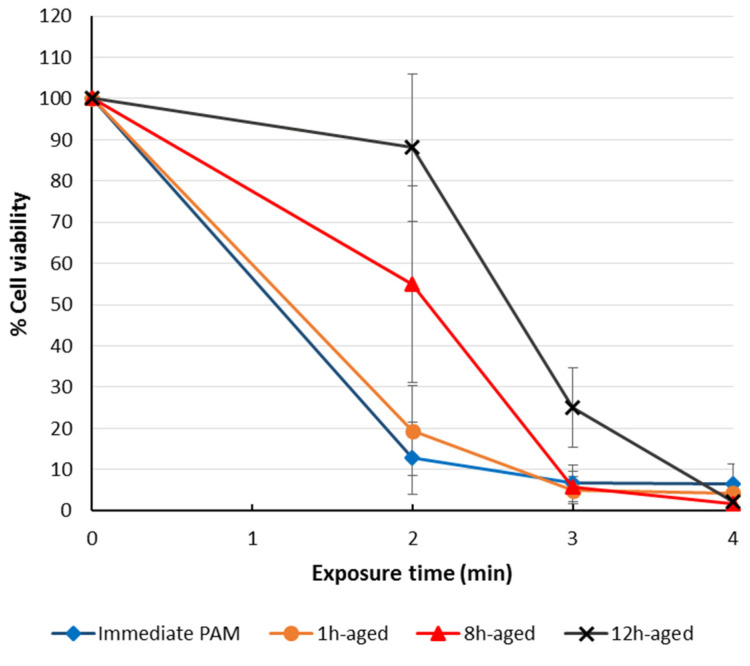
Effectiveness of stored/aged PTS (cell culture medium) to induce cell death in SCaBER cells for storage times of 1, 8, and 12 h, when measuring metabolic activity another 12 h later. Reproduced from [82].

**Figure 11 cancers-13-01737-f011:**
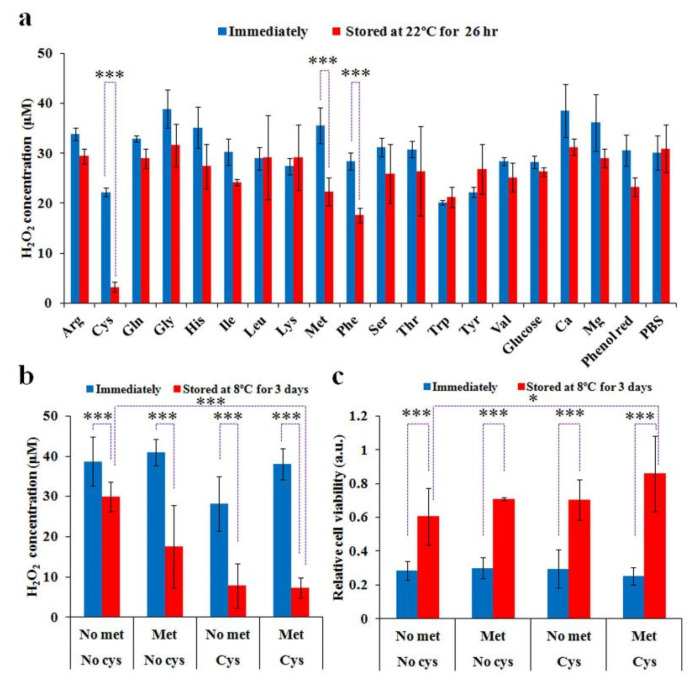
Cysteine and methionine mainly cause the degradation of PTS (cell culture medium) at 22°C and 8 °C. (**a**) The H_2_O_2_ concentration in PTS (PBS) containing a specific component during the storage at 22 °C. (**b**) The H_2_O_2_ concentration in the PTS (cys/met-free DMEM, cys-free DMEM, met-free DMEM, and standard DMEM) during the storage at 8 °C. (**c**) The anticancer effect of the PTS (cys/met-free DMEM, cys-free DMEM, met-free DMEM, and standard DMEM) after storage at 8 °C. Student’s *t*-tests were performed and the significance is indicated as * *p* < 0.05 and *** *p* < 0.005. Reproduced from [31]. Copyright 2016 Springer Nature.

**Figure 12 cancers-13-01737-f012:**
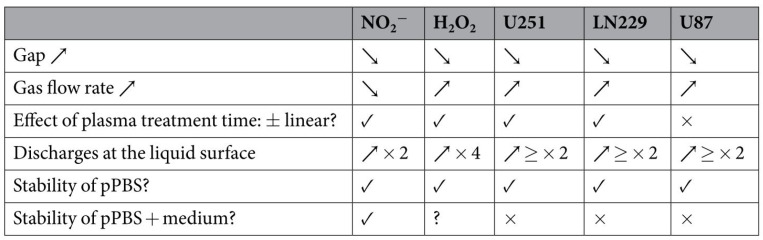
The effect of the gap, gas flow rate, plasma treatment time, the occurrence of discharges at the liquid surface, and the stability of a PTS (PBS, kINPen argon plasma jet) on the concentrations of NO_2_^-^ and H_2_O_2_ in PTS and on the anticancer capacity of PTSs for three different cancer cell lines. Reproduced from [85]. Copyright 2017 Springer Nature.

**Figure 13 cancers-13-01737-f013:**
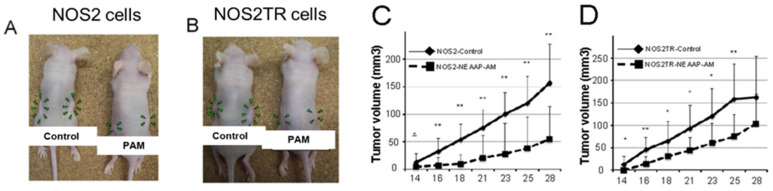
Antitumor effect of PTS (RPMI, “PAM”) in mice with NOS2 and NOS2TR (paclitaxel-resistant) cell lines. (**A**,**B**): The macroscopic observation of nude mice bearing subcutaneous NOS2 (**A**) and NOS2TR (**B**) tumors on both flanks. Mice were injected with NOS2 and NOS2TR cells and then received medium alone or NEAPP-AM. A total of 0.4 mL of medium or NEAPP-AM was administered locally into both sides of mice three times a week. All mice were sacrificed at 29 days after implantation. Green arrowheads indicate tumor formation. (**C**,**D**): Time-dependent changes in the tumor volume in xenografted models are shown, medium alone or NEAPP-AM. Each point on the line graph represents the mean tumor volume (mm^3^) for each group on a particular day after implantation, and the bars represent SD. * *p* < 0.05, ** *p* < 0.01 versus control. Reproduced from [39]. Copyright 2013 PLOS.

**Figure 14 cancers-13-01737-f014:**
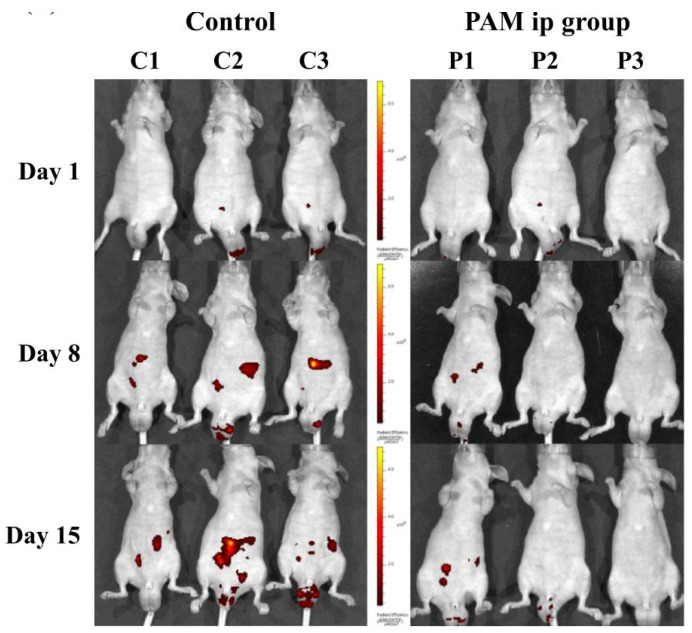
Efficacy of the intraperitoneal administration of PTS (“PAM”) in vivo. Reproduced from [78]. Copyright 2017 Springer Nature.

**Figure 15 cancers-13-01737-f015:**
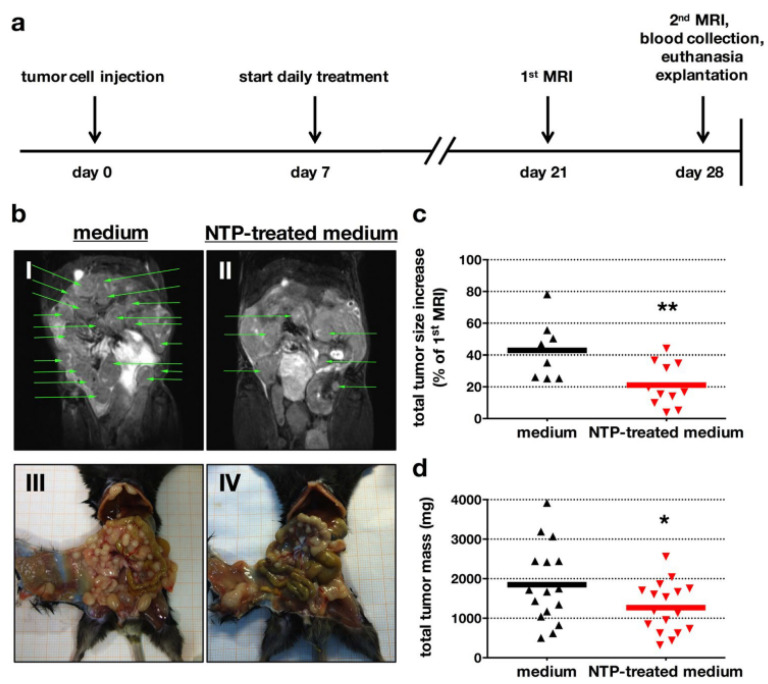
Efficacy of the repeated intraperitoneal administration (**a**) of PTS (RPMI medium without supplements) against syngeneic, orthotopic, disseminated pancreatic cancer analyzed using MR-Imaging (I–II) and macroscopic imaging (III–IV) (**b**) and its quantification (**c**) as well the absolute tumor weights (**d**). Each triangle represents one mouse. * *p* < 0.05, ** *p* < 0.01. Reprinted with permission from [86]. Copyright 2017 Springer Nature.

**Figure 16 cancers-13-01737-f016:**
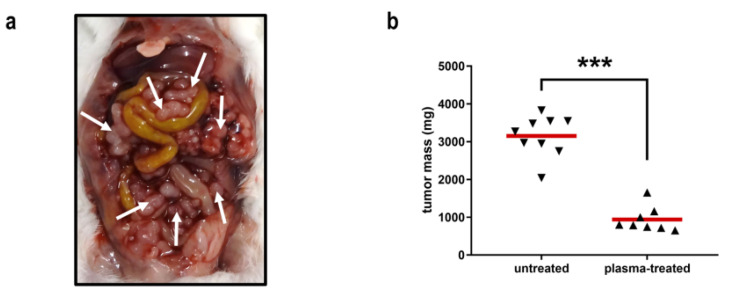
Efficacy of the repeated intraperitoneal administration of PTS (NaCl, kINPen argon plasma jet) against syngeneic, orthotopic and disseminated colorectal in mice (**a**) and its absolute tumor weights (**b**). Each triangle represents one mouse. *** *p* < 0.001. Reproduced from [40]. Copyright 2019 Springer Nature.

**Table 1 cancers-13-01737-t001:** Chemical composition of six clinically approved solutions compared to phosphate-buffered saline (PBS) and RPMI cell culture medium containing 10% fetal bovine/calf serum (R10F). Reproduced from [37]. Copyright 2019 IEEE.

	HES	NaCl	G-5	E153	Ri-Lac	Gela	PBS	R10F
main component	60 g/L hydroxyethyl starch	9 g/500 mL sodium chloride	50 g/L glucose	153 mval/L ions	Ringer’s solution with 28 mmol/L	Gelatine 40 g/L	12 mM phosphate	Amino acids, vitamins and 10% FCS
pH-range	4.0–5.5	4.5–7.0	3.5–5.5	5.0–7.0	5.0–7.0	7.1–7.7	7.3–7.5	8.0
pH-treated	5.2	5.1	5.6	6.2	6.0	7.0	7.3	8.3
osmolarity (mOsm)	308	308	278	303	277	274	280	-
acetions				X				
amino acids								X
Ca				X	X		X	
calcium hydrochloride-dihydrate				X	X			
calcium nitrate								X
carbohydrates	X		X					X
Cl					X			
gelatine poly succinate						X		
HCl	X							
K					X			
KCl				X	X		X	
lactate					X			
magnesium sulfate								X
magnesium chlorid-hecyhydrat				X				
Mg				X				
NaCl	X	X		X	X	X	X	X
phosphate							X	X
protein								X
sodium acetate				X	X			
sodium hydroxide	X							
vitamins								X

## Data Availability

Not applicable.

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
