# Peer review of "Plasma-Treated Solutions (PTS) in Cancer Therapy"

_cancers, 2021, doi:10.3390/cancers13071737_

Round 1
Reviewer 1 Report
The manuscript entitled “Plasma-Treated Solutions (PTS) in Cancer Therapy” by Liu et al. discussed the anticancer effect of PTS/PAM/PAS/PAL/PTL/PSS and importance of large volume of sterile PTS. This review addresses the challenges and opportunities of plasma-treated solutions (PTS) for cancer treatment. These results mentioned in this review manuscripts are collectively vital in understanding the plasma liquid-chemistry for use in oncology and will help design more optimal liquids for further exploration in research and future applications. This review manuscript is nice and need minor revision. The issues are listed as below.
- Author must describe direct and indirect plasma treatment based on plasma physics view-point also in Section 1 or 2.
- Author must improve quality of Table 1 as many words are not readable.
- I recommend author to improve Figure 2. Try to include more species and factors generated by plasma on surface, in gas and liquids.
- DAF is not only for NO, try to make it NOx or RNS.
- Author must describe previous studies and reviews based on plasma-liquid for biological applications, especially for cancer treatment. An example of a reference is given here. (Kaushik N K, Ghimire B, Li Y, Adhikari M, Veerana, M, Kaushik N, Jha N, Adhikari B, Lee SJ, Masur M, Woedtke Tv, Weltmann K D, Choi E H. 2018 Biol. Chem. 400 039062)
Reviewer 2 Report
Low-temperature plasma and plasma treated solutions brings represents hope in oncology therapy. This manuscript brings review of plasma-treated solutions from their definition through their affects, efficiancy and mechanisms researched in in vitro and in vivo experiments. The findings are also suitably demostrated by graphs and figures from previous publications, which increases quality of the manuscript.
I have some comment on the formal page:
- p4, line 149: there is an extra parenthesis
I have some questions:
- You have mentioned immuno-modulatory properties of PTS. Could you specify them in more details? Could be PTS used in combination with immuno therapy? What is the effect of PTS application on T lymphocytes infiltrating tumour?
- Every tumor is specific and has its own microenvironment, deregulated pathways and mutational load. How these factors could influence sensitivity of cells to ROS and efficiency of PTS? Should the use of PTS be personalized based on molecular characterization of tumor tissue?
- Different tumor cell lines showed different sensitivity to certain types of PTS. What does the effectiveness of specific types of PTS depend on?
I think it would be appropriate to add these answers to the text.
Reviewer 3 Report
This impressive study has been very carefully performed by some of the most internationally reputed searchers in the field. The study is comprehensive and focuses on the interest of plasma treated solutions in oncology, compared with direct plasma application.
Important points for future works and a rapid clinical translation are very clearly developed. The need for a standardization of methods is particularly highlighted, encouraging authors of reports to give « the exact chemical composition of the type of media » as well as « the essential types of ROS/RNS generated in the solution by the plasma source », two essential points allowing a better comparison between studies.
This review will be of great value for students, basic scientists as well as physicians, inside and outside the field of plasma medicine. The main short lived reactive species in the plasma phase and long lived reactive species in the solution are reported, together with their possible influence and limitations upon cancer cells and tumors, as observed in in vitro and in vivo experiments. Some intracellular mechanisms involved in cancer cell death or tumor size reduction are also presented.
Finally, if a hormetic cell response to PTS can be assumed (which is, from my opinion, still questionable), studies similar to the one of Xu et al. (ref [34]) presented in paragraph 4.2. have to be performed in order to identify thresholds (RONS concentrations, exposure times, …) inducing PTS toxic effects. Similarly, the selectivity of PTS treatments on cancer cells should also be evoked in Part 3 concerning in vitro experiments.
Nevertheless, I highly recommend a rapid publication.
